# Microscopic vascular invasion may not be associated with survival of patients undergoing resection for solitary hepatoma of ≤ 2 cm

Wei-Feng Li[1], Yueh-Wei Liu[1], Chih-Chi Wang[1☯]*, Chee-Chien Yong[1], Chih-Che Lin[1], Yi-Hao Yen[2☯]*

1 Liver Transplantation Center and Department of Surgery, Kaohsiung Chang Gung Memorial Hospital, Kaohsiung, Taiwan, 2 Division of Hepatogastroenterology, Department of Internal Medicine, Kaohsiung Chang Gung Memorial Hospital and Chang Gung University College of Medicine, Kaohsiung, Taiwan

☯ These authors contributed equally to this work.
* ufel4996@ms26.hinet.net (CCW); cassellyen@yahoo.com.tw (YHY)

## Abstract

### Background/objective

To determine the impact of microvascular invasion (MVI) on outcome in patients with solitary hepatocellular carcinoma (HCC) of ≤ 2 cm undergoing liver resection (LR).

### Methods

This retrospective study enrolled consecutive patients between 2007–2019 with newly diagnosed solitary HCC ≤ 2 cm who were undergoing LR at our institution. Overall survival (OS) and recurrent-free survival (RFS) were compared between patients with or without MVI.

### Results

Of the 229 patients included in this study, 71 had MVI. The median follow-up period was 28.8 months (interquartile range: 13.5–70.1). Although the 90-day mortality rate was 0, 18 deaths occurred during the study, and the 5-year survival rate was 87.1%. Tumor recurrence occurred in 45 cases, and 5-year RFS was 71.9%. The presence or absence of MVI did not significantly affect the OS and RFS rates (log rank test, p = 0.10 and 0.38, respectively). In univariate and multivariate analysis, the presence of MVI was not associated with OS and RFS.

### Conclusion

The presence of MVI was not associated with OS and RFS in patients with solitary HCC ≤ 2 cm who underwent LR in this cohort.

**Data Availability Statement:** The raw data of the present cohort are available at the following DOI: https://data.mendeley.com/drafts/63dzpdf37r.

**Funding:** This study was supported by Grant CMRPG8L0181 from the Chang Gung Memorial Hospital-Kaohsiung Medical Center, Taiwan. The funders had no role in study design, data collection and analysis, decision to publish, or preparation of the manuscript.

**Competing interests:** The authors have declared that no competing interests exist.

## Introduction

The American Joint Committee on Cancer (AJCC) tumor-node-metastasis (TNM) staging system is a commonly used system for patients with hepatocellular carcinoma (HCC) who undergo liver resection (LR) [1, 2]. In the most recent version of the AJCC staging system [1], solitary tumors ≤ 2 cm with or without microscopic vascular invasion (MVI) are categorized as T1a, based on the results of an international multi-institutional study [3]. This study enrolled 155 patients with solitary HCC ≦ 2 cm who underwent LR. Of these, 41 patients had MVI, and no significant difference was found in OS between patients with or without MVI (log rank test, p = 0.8) [3].

However, MVI is a strong predictor of overall survival (OS) and recurrent-free survival (RFS) following LR of HCC [4, 5]. The breakdown of extracellular matrix, loss of cell-to-cell adhesion, and use of cellular motility and alternative energy are required for MVI [6]. Presence of MVI can also indicate aggressive tumor biological behavior [6].

Several studies have shown that MVI does not influence OS in patients with solitary HCC ≦ 2 cm who underwent LR [3, 7–12]. Of these studies, three [8, 11, 12] enrolled patients who were found to have satellite nodules on pathology, the presence of which would be regarded as multiple tumors based on TNM classification [1, 2]. In addition to these studies [8, 11, 12], Yamashita, et al. evaluated the impact of microinvasion on the prognosis of patients with solitary HCC ≦ 2 cm who underwent LR. Microinvasion is defined as a composite of MVI, bile duct invasion, and intrahepatic metastasis (i.e. satellite nodules) [10]. Base on the results of these studies, it was difficult to determine whether MVI impacts treatment outcomes in patients with solitary HCC ≤ 2 cm who underwent LR [8, 10–12].

A previous study using the Surveillance, Epidemiology and End Results (SEER) database enrolled patients undergoing LR or liver transplant (LT) for HCC with a pathological tumor grade of T1a. 111 of the patients had MVI, while 957 did not, and no significant difference was found in OS between the two groups (p = 0.5) [9]. Another study that used the SEER database to enroll patients undergoing LR for HCC also found that the OS was not significantly different between patients with solitary HCC ≦ 2cm with or without MVI (MVI–, n = 385; MVI+, n = 54; p = 0.0693) [7]. The SEER database provides the advantage of a large sample size; however, it lacks details of clinical, operative, and pathologic data, which are important for prognosticating the outcomes of HCC patients undergoing LR.

Relatedly, the aim of this study was to clarify whether or not MVI has an impact on OS and RFS in patients with solitary HCC ≤ 2 cm undergoing LR at a single institution using granular review of medical records.

## Methods

The Institutional Review Board of Kaohsiung Chang Gung Memorial Hospital approved this study (IRB number:202000398B0). The requirement for informed consent was waived by the IRB due to the retrospective and observational nature of this study. The data used were extracted from the Kaohsiung Chang Gung Memorial Hospital HCC registry database. The data in the database were prospectively collected. We checked the vital statuses of the patients annually in the database using the Cancers Screening and Tracing Information Integrated System for Taiwan (https://hosplab.hpa.gov.tw/CSTIIS/index.aspx).

The algorithm of the patient enrollment from 2007–2017 is shown in S1 Fig. There were 1083 patients with newly diagnosed HCC who underwent LR between 2007–2017. Of these, 172 (15.9%) were pathologically proven to have solitary HCC ≦ 2 cm. The 8th AJCC TNM was applied from 2018 onwards. There were 199 patients with newly diagnosed HCC undergoing LR in 2018. Of these, 26 (13.1%) were graded pathologically as T1aN0M0. There were 197

patients with newly diagnosed HCC undergoing LR in 2019, of which 31 (15.7%) were graded pathologically as T1aN0M0. A total of 229 patients with pathologically graded T1aN0M0 [1] HCC were therefore enrolled in this study.

The raw data of the present cohort are available at the following DOI: https://data.mendeley.com/drafts/63dzpdf37r.

Clinical, pathological, and operative data were collected for analysis using medical records reviewed by Y.H.Y. Clinical data included age, sex, etiology of chronic liver disease, preoperative platelet count, albumin, total bilirubin, aspartate aminotransferase (AST), alanine aminotransferase (ALT), creatinine, international normalized ratio (INR), Child Pugh class [13], model for end-stage liver disease (MELD) score [14], albumin-bilirubin (ALBI) grade [15], α-feto protein (AFP), results of indocyanine green retention test at 15 minutes (ICG-R15), and antiviral therapies for chronic hepatitis B virus (HBV) and hepatitis C virus (HCV) infection. Operative data collected included the extent of resection (major vs minor), the approach for resection (open vs laparoscopic), operative duration, and blood loss during surgery. Anatomic resection was defined as removal of the entire Couinaud segment(s) involved with the tumor [8]. Major resection was defined as removal of ≧ 3 Couinaud segments. Post-operative complications were assessed and graded according to the Clavien-Dindo classification, with severe post-operative complications defined as grade ≧ III [16]. The pathological data collected were as follows: The size of each tumor was determined according to the dimensions of the tumor in the resected specimen. MVI was defined as tumor thrombi within a vascular space lined by endothelium that were visible only on microscopy [17]. Tumor grade was assessed according to Edmondson and Steiner classification [18]. Liver fibrosis was assessed according to the Ishak score, with scores of 5 or 6 considered to be indicative of cirrhosis [19]. OS was calculated from the date of LR to the date of death (event), or to the date of last follow-up visit (censored). RFS was calculated from the date of LR to the date of HCC recurrence (event), or to the date of last follow-up visit (censored). Patients undergoing LT were censored at the time of transplant.

The general principle of surveillance following LR at Kaohsiung Chang Gung Memorial Hospital adheres to the following guideline recommendations: computed tomography or magnetic resonance imaging every 3–6 months for 2 years, and annually thereafter; AFP levels and ultrasound every 3 months for 2 years, and every 6 months thereafter [20–22]. The diagnosis of recurrence was based on international guidelines [20–22] and/or multidisciplinary team discussions.

## Treatment of recurrent tumors

In general, for patients with recurrent tumors that meet the Milan criteria [23] and have a tumor size of ≦ 3.0 cm, radiofrequency ablation (RFA) is indicated. For patients with a solitary recurrent tumor ≦ 2.0 cm, percutaneous ethanol injection (PEI) may be considered as an alternative treatment to RFA. Repeated LR is indicated for patients with a solitary recurrent tumor who have adequate remaining liver volume and function reserve. For patients with recurrent tumors within the Milan criteria [23] and inadequate liver function reserve (i.e., decompensated cirrhosis or clinically significant portal hypertension (CSPH)), LT is indicated. CSPH is defined by the presence of portosystemic collateral vessels, or a platelet count $< 10^9$ /L and splenomegaly [24]. Transarterial chemoembolization (TACE) is indicated for patients with multiple recurrent tumors beyond the Milan criteria [23]. Sorafenib may be used in patients with macrovascular invasion or extrahepatic metastasis [20–22].

## Antiviral therapies

Antiviral therapies with nucleos(t)ide analogues for chronic hepatitis B are indicated according to the following guideline recommendations: (1) chronic hepatitis B patients with persistently

elevated ALT and HBV deoxyribonucleic acid (DNA); or (2) cirrhotic patients or patients who underwent curative resection for HCC with elevated HBV DNA, irrespective of ALT levels [25, 26]. Chronic hepatitis C antiviral treatment used pegylated interferon combined ribavirin in 2007–2017, and direct-acting antiviral agents since 2017. These antiviral therapies were indicated for patients who were anti-HCV positive and viremic [27, 28].

## Statistical analysis

The characteristics of the patients were reported as medians and interquartile range (IQR) for continuous variables, while the categorical variables were reported as frequencies and percentages. A Chi-square test was used to compare categorical variables within groups. The Mann–Whitney U test was used to compare continuous variables within groups. OS and RFS were compared within groups using the Kaplan-Meier method and log-rank test. To identify risk factors for OS and RFS, a multivariate regression analysis was performed using the Cox proportional hazards model for variables with p values < 0.05 in the univariate analysis. All *P* values were two-sided, and the significance level was set at 0.05. All statistical analyses were performed using the MedCalc® (version 20.015; MedCalc Software, Ostend, Belgium; http://www.medcalc.org).

## Results

### Clinical, operative, and pathological characteristics

229 patients were enrolled in this study. The clinical data were as follows: Of 229 patients, 62 (27.1%) were > 65 years old, 163 (71.2%) were male, 79 (34.5%) had AFP > 20 ng/ml, 224 (97.8%) were Child Pugh class A, 123 (53.7%) were hepatitis B surface antigen (HBsAg) positive, 86 (37.6%) were anti-HCV positive, and 8 (3.5%) were alcohol abusers. The median laboratory values for these patients were: AST 109 U/L (IQR: 43–207), ALT 118 U/L (IQR: 45–197), albumin 3.6 g/dL (IQR: 3.2–3.9), creatinine 0.8 mg/dL (IQR: 0.7–1.0), total bilirubin 1.0 mg/dL (IQR:07–1.4), INR 1.1 (IQR: 1.1–1.2), ICG-R15 8.9% (IQR: 5.8–14.5), and platelet count 139 $10^9$/L (IQR:112–179).

The operative data were as follows: Of 229 patients, 64 (27.9%) underwent major resection, 72 (31.4%) underwent laparoscopic resection, 161 (70.3%) underwent anatomic resection, and 115 (50.2%) had blood loss of $\leqq$ 100 ml during the operation. The median surgery duration was 4.5 hours (IQR: 4–5).

The pathological data were as follows: Of 229 patients, 71 (31.3%) had MVI, 36 (15.7%) had well-differentiated tumors, 185 (80.8%) had moderately-differentiated tumors, and 5 (2.2%) patients had poorly-differentiated tumors. 125 (54.6%) patients were cirrhotic, and the median tumor size was 16 mm (IQR: 15–19). R0 resection was noted in all patients.

No significant differences in these characteristics were found between patients with MVI, and those without (Table 1).

### Survival analysis

**All patients.** The median follow-up period was 28.8 months (IQR: 13.5–70.1). The 90-day mortality rate was 0. 18 deaths occurred, and 5-year survival was 87.1%. Of the 5 patients who underwent LT, all were alive at the last follow-up visit. 45 patients suffered tumor recurrence, and 5-year RFS was 71.9%. There was no significantly difference in OS between patients with and without MVI (log rank test, p = 0.10) (Fig 1). Univariate analysis showed no variable associated with OS (Table 2).

**Table 1. Characteristics of patients with HCC up to 2 cm by microvascular invasion status.**

| Characteristic | All patients, n = 229 | MVI- (n = 158) | MVI+ (n = 71) | p |
|---|---|---|---|---|
| Age > 65 years | 62 (27.1%) | 43 (27.2%) | 19 (26.8%) | 0.943 |
| Male | 163 (71.2%) | 110 (69.6%) | 53 (74.6%) | 0.437 |
| AFP >20 ng/ml | 79 (34.5%) | 51 (32.3%) | 28 (39.4%) | 0.292 |
| HBsAg positive | 123 (53.7%) | 84 (53.2%) | 39 (54.9%) | 0.804 |
| Anti-HCV positive | 86 (37.6%) | 59 (37.3%) | 27 (38.0%) | 0.921 |
| Alcohol abuse | 8 (3.5%) | 4(2.5%) | 4 (5.6%) | 0.258 |
| AST (U/L) [a] | 109 (43–207) | 109 (48–209) | 108 (41–205) | 0.433 |
| ALT (U/L) [b] | 118 (45–197) | 119 (46–222) | 112 (40–185) | 0.378 |
| Albumin (g/dL) [c] | 3.6 (3.2–3.9) | 3.6 (3.2–4.0) | 3.6 (3.25–3.9) | 0.981 |
| ICG-R15 (%) [d] | 8.9 (5.8–14.5) | 9.7 (5.9–14.6) | 8.3 (5.3–14.3) | 0.446 |
| Platelet ($10^9$/L) [e] | 139 (112–179) | 139 (112–179) | 138 (114–181) | 0.599 |
| Creatinine (mg/dL) [f] | 0.8 (0.7–1.0) | 0.8 (0.7–1.0) | 0.8 (0.7–1.0) | 0.817 |
| Total bilirubin (mg/dL) | 1.0 (07–1.4) | 1.0 (0.7–1.5) | 1.1 (0.7–1.4) | 0.789 |
| INR [g] | 1.1 (1.1–1.2) | 1.1 (1.1–1.2) | 1.1 (1.1–1.2) | 0.288 |
| Child Pugh class A | 224 (97.8%) | 154 (97.5%) | 70 (98.6%) | 0.99 |
| MELD score [h] | 5.5 (3.6–7.9) | 5.3 (3.5–8.3) | 5.8 (4.3–7.5) | 0.654 |
| ALBI score [i] | | | | 0.913 |
| Grade 1 | 49 (22.3%) | 35 (23.2%) | 14 (20.3%) | |
| Grade 2 | 164 (74.5%) | 111 (73.5%) | 53 (76.8%) | |
| Grade 3 | 7 (3.2%) | 5 (3.3%) | 2 (2.9%) | |
| Laparoscopic resection | 72 (31.4%) | 48 (30.4%) | 24 (33.8%) | 0.606 |
| Anatomic resection | 161 (70.3%) | 111 (70.3%) | 50 (70.4%) | 0.979 |
| Blood loss ≦100ml | 115 (50.2%) | 80 (50.6%) | 35 (49.3%) | 0.852 |
| Operative duration (hours) | 4.5 (4–5) | 5 (4–5) | 4 (4–5) | 0.315 |
| Major resection | 64 (27.9%) | 48 (30.4%) | 16 (22.5%) | 0.221 |
| Tumor size (mm) | 16 (15–19) | 16 (14–19) | 17 (15–18) | 0.178 |
| Tumor differentiation | | | | 0.054 |
| Well | 36 (15.7%) | 30 (19.0%) | 6 (8.5%) | |
| Moderate | 185 (80.8%) | 123 (77.8%) | 62 (87.3%) | |
| Poor | 5 (2.2%) | 2 (1.3%) | 3 (4.2%) | |
| Necrosis | 3 (1.3%) | 3 (1.9%) | 0 | |
| Cirrhosis | 125 (54.6%) | 83 (52.5%) | 42 (59.2%) | 0.352 |
| R0 resection | 229 (100%) | 158 (100%) | 71 (100%) | 0.99 |

MVI, microscopic vascular invasion; AST, Aspartate aminotransferase; ALT, Alanine aminotransferase; INR, international normalized ratio; AFP, α-feto protein; ICG-R15, indocyanine green retention test at 15 minutes; HBsAg, hepatitis B surface antigen; HCV, hepatitis C virus; MELD, Mayo End-Stage Liver Disease; ALBI, Albumin-Bilirubin.

Number of missing data: a, n = 13; b, n = 13; c, n = 6; d, n = 33; e, n = 6; f, n = 1; g, n = 2; h, n = 9, i = 9

There was no significant difference in the RFS between those with and without MVI (log rank test, p = 0.38) (Fig 2). Univariate analysis showed that HBsAg positive status (hazard ratio (HR) = 0.498; 95% confidence interval (CI) = 0.277–0.895; p = 0.02), poor differentiation of tumor cells (HR = 4.311; 95% CI = 1.325–14.028; p = 0.015), and creatinine levels of > 1.2 mg/dl (HR = 2.269; 95% CI = 1.177–4.374; p = 0.014) were associated with RFS. However, these associations were found to be insignificant on multivariate analysis (Table 3).

**Severe post-operative complications.** Three patients suffered severe post-operative complications [16]. One developed post-hepatectomy liver decompensation with ascites and

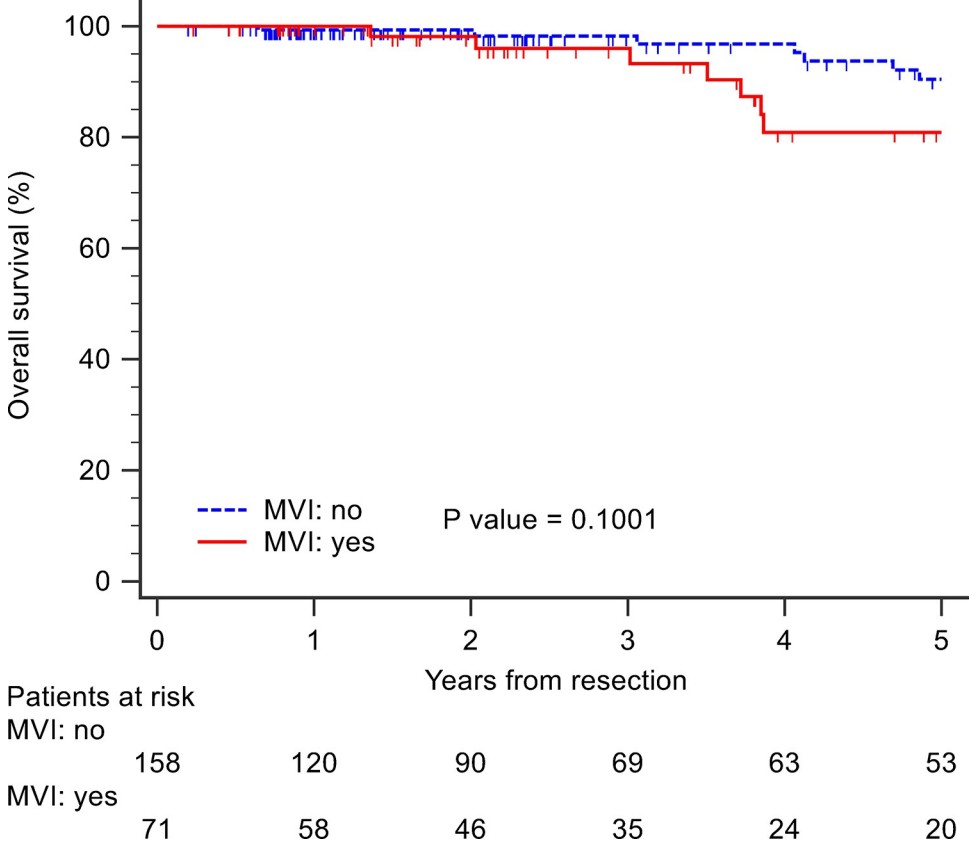

**Fig 1. Overall survival of patients with solitary hepatocellular carcinoma according to presence of microvascular invasion (MVI).**

hypoalbuminemia resulting in a right pleural effusion, and pig-tail drainage was performed. Pneumothorax was later noted (for which a chest tube was inserted), as well as acute kidney injury and hyponatremia. Eventually, full recovery to compensated liver function was observed in this patient. Another patient developed wound dehiscence, and underwent suturing of the wound 3 days following LR. The third patient developed a right pleural effusion, and pigtail drainage was performed. No patient died from the operation.

## Subgroup analysis

**Prognostic impact of MVI.**   When comparing the outcomes of patients with and without MVI, no significant difference in OS and RFS was observed in cirrhotic patients (p = 0.26 and 0.63, respectively) (S2 and S3 Figs). No significant difference in OS and RFS was also observed in non-cirrhotic patients (p = 0.15 and 0.58, respectively) (S4 and S5 Figs), in female patients (p = 0.10 and 0.08, respectively) (S6 and S7 Figs), in male patients (p = 0.37 and 0.97, respectively) (S8 and S9 Figs); in HBsAg positive patients (p = 0.38 and 0.29, respectively) (S10 and S11 Figs); or in anti-HCV positive patients (p = 0.21 and 0.81, respectively) (S12 and S13 Figs).

**Prognostic impact of antiviral therapies.**   When comparing the outcome of patients given or not given antiviral therapies, there was no significant difference in OS and RFS in HBsAg positive patients (p = 0.80 and 0.61, respectively) (S14 and S15 Figs) or in anti-HCV positive patients (p = 0.99 and 0.66, respectively) (S16 and S17 Figs).

**Table 2. Univariate analysis of overall survival.**

| | Univariable | |
| --- | --- | --- |
| | HR (95%CI) | P |
| Age (years) | | |
| >65 | 1.652 (0.572–4.766) | 0.353 |
| ≤65 | Reference | |
| Sex | | |
| Male | 2.302 (0.908–5.836) | 0.079 |
| Female | Reference | |
| Anti-HCV | | |
| Positive | 1.240 (0.480–3.204) | 0.657 |
| Negative | Reference | |
| HBsAg | | |
| Positive | 0.395 (0.148–1.055) | 0.064 |
| Negative | Reference | |
| AFP (ng/ml) | | |
| >20 | 1.031 (0.399–2.665) | 0.950 |
| <20 | Reference | |
| Creatinine (mg/dL) | | |
| >1.2 | 1.647 (0.536–5.061) | 0.384 |
| ≦1.2 | Reference | |
| Child Pugh class | | |
| B | 0.042 (0.000–259.909) | 0.476 |
| A | Reference | |
| Platelet count ($10^9$/L) | | |
| <150 | 1.925(0.536–6.911) | 0.316 |
| ≧150 | reference | |
| Albumin (g/dL) per 1 increase | 0.614(0.212–1.775) | 0.368 |
| MELD score ≧9 | 1.278(0.355–4.604) | 0.707 |
| MELD score <9 | reference | |
| ALBI grade 2 or 3 | 2.630(0.343–20.143) | 0.352 |
| ALBI grade 1 | Reference | |
| Extent of resection | | |
| Major | 0.592 (0.194–1.809) | 0.358 |
| Minor | Reference | |
| The approach of resection | | |
| Open | 0.735 (0.156–3.461) | 0.697 |
| Laparoscopic | Reference | |
| Anatomic resection | | |
| No | 1.800 (0.697–4.649) | 0.225 |
| Yes | Reference | |
| Blood loss during operation | | |
| >100 ml | 0.493 (0.185–1.317) | 0.158 |
| ≦100 ml | Reference | |
| Operative duration (hours) Per 1 hour increase | 0.741 (0.510–1.077) | 0.116 |
| Microvascular invasion | | |
| Yes | 2.195 (0.840–5.735) | 0.109 |
| No | Reference | |
| Cirrhosis | | |

(*Continued*)

**Table 2.** (Continued)

| | Univariable | |
|---|---|---|
| | **HR (95%CI)** | **P** |
| Yes | 1.445 (0.541–3.856) | 0.463 |
| No | Reference | |
| Tumor differentiation | | |
| Poor | 5.204 (0.674–40.172) | 0.114 |
| Well or moderate | Reference | |

MVI, microscopic vascular invasion; AFP, α-feto protein; HBsAg, hepatitis B surface antigen; HCV, hepatitis C virus; MELD, Mayo End-Stage Liver Disease; ALBI, Albumin-Bilirubin

Of the 45 patients who suffered tumor recurrence, 30 did not have MVI. 26 (86.7%) of these 30 patients were within the Milan criteria [23]. 14 (93.3%) of the 15 patients who suffered tumor recurrence and had MVI were also within the Milan criteria [23]. The proportion of patients with recurrent tumors within the Milan criteria [23] was not found to be significantly different (p = 0.65) between the group with MVI and the group without MVI. Of the 30 patients with tumor recurrence who did not have MVI, 3 (10.0%) underwent repeated LR, 12 (40.0%) underwent RFA, 1 (3.3%) underwent LT, 11 (36.7%) underwent TACE, 1 (3.3%) underwent radiation therapy for a hilar tumor with bile duct invasion, and 2 (6.7%) patients were lost to follow-up. Of the 15 patients with tumor recurrence who had MVI, 2 (13.3%) underwent LT, 2 (13.3%) underwent PEI, 1 (6.7%) underwent repeated LR, 7 (46.7%) underwent RFA, 1 (6.7%) underwent radiation and sorafenib therapy for brain and lung metastases, and 2 (13.3%) underwent TACE. 16 (53.3%) patients without MVI underwent curative treatments (i.e., LT, LR or RFA), compared to 12 (80.0%) patients with MVI (p = 0.08).

## Discussion

In this study, we analyzed the influence of MVI on the OS and RFS of 229 patients with solitary HCC ≤ 2 cm who underwent LR. Our findings indicate that the presence of MVI was not associated with worse OS and RFS in these patients with HCC ≤ 2 cm who underwent LR. The rates of OS and RFS at 5 years were excellent irrespective of MVI status, suggesting that the 8th AJCC TNM classification of T1a is appropriate. Liver function reserve may play a pivotal role in the prognosis of patients with very early-stage tumors. Furthermore, subgroup analysis showed that the MVI status did not impact OS and RFS in patients from HBsAg positive, anti-HCV positive, cirrhotic, or non-cirrhotic subgroups. The Child-Pugh score [13] is widely used for assessment of liver function reserve and prediction of postoperative outcomes in patients with HCC [20–22]. In addition to the Child-Pugh score, a previous study reported that a MELD score of ≧ 9 was an independent predictor of worse OS in HCC patients who underwent LR [29]. Another study reported that a high ALBI grade was an independent predictor of early HCC recurrence following LR [30]. Our study found that none of these 3 methods of assessing liver function reserve were associated with OS or RFS.

In addition, previous studies have reported that antiviral therapy following curative therapies (i.e. LR or RFA) reduces HCC recurrence rates [31, 32]. However, in our cohort, anti-viral therapies were not found to have an impact on OS and RFS in HBV and HCV-infected patients. The discrepancy between the results of our study and those of previous studies [31, 32] could be due to differences in the patients' tumor stages. While previous studies enrolled HCC patients who had undergone LR irrespective of tumor stage [31, 32], our study enrolled only patients with earliest-stage HCC, and found that the OS and RFS were excellent irrespective of antiviral therapy.

Our study did not identify any variables significantly associated with OS in patients with pathological T1a tumors who underwent LR, which is consistent with results of the previous

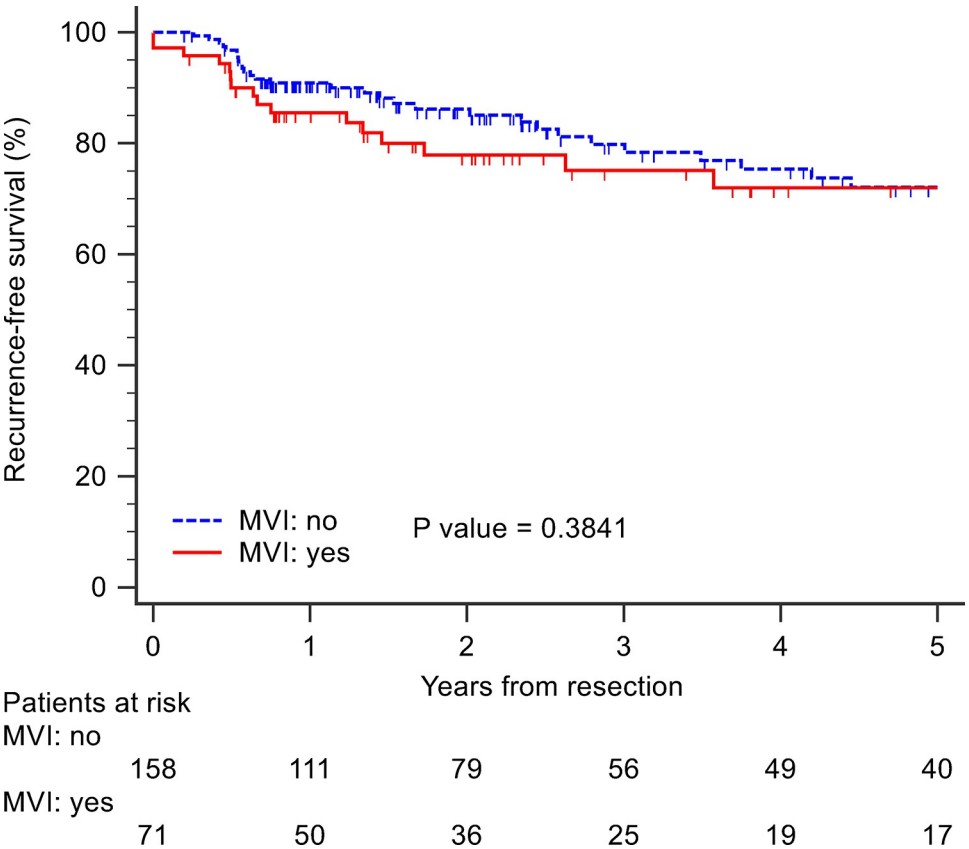

**Fig 2. Recurrent-free survival rates of patients with solitary hepatocellular carcinoma according to presence of microvascular invasion (MVI).**

study [3]. Furthermore, MVI was not found to impact OS in these patients, which is also consistent with results of previous studies [3, 7, 9].

Only 125 (54.6%) of the patients in this study were cirrhotic. Our previous study showed that cirrhosis is found in 74% of patients with newly diagnosis HCC (patients who underwent surgical and non-surgical treatments were enrolled) [33]. However, the prevalence of cirrhosis in the current study is consistent with previous studies that enrolled patients with solitary HCC ≦ 2 cm who underwent LR [3, 8, 10, 11]. Previous studies from Taiwan and Japan reported that 53.5% and 55% of patients had cirrhosis, respectively [10, 11], compared to 67% of patients in a previous study from Western countries [8]. An international study from Japan and several Western countries reported that 60% patients had cirrhosis [3]. As only patients with adequate liver function reserve are selected to undergo LR, this could explain the lower prevalence of cirrhosis observed in HCC patients selected for LR, compared to patients who underwent surgical or non-surgical treatments [33].

The strengths of our study were as follows: Firstly, our study used fine granular review of medical records to enroll the largest case number of patients with pathologically proven solitary HCC ≦ 2 cm who underwent LR. By collecting details of clinical, pathological, and operative prognostic factors, we were able to evaluate whether MVI impacts patient outcome. Secondly, the previous study did not evaluate the impact of MVI on RFS in patients with solitary HCC ≦ 2 cm undergoing LR [3]. In contrast, this study found that the presence of MVI was not associated with reduced RFS in these patients. Thirdly, we checked the vital status of

**Table 3. Univariate and multivariate analysis of recurrent free survival.**

| | Univariable | | Multivariable | |
|---|---|---|---|---|
| | HR (95%CI) | P | HR (95%CI) | P |
| Age (years) | | | | |
| >65 | 1.575 (0.850–2.919) | 0.149 | | |
| ≤65 | Reference | | | |
| Sex | | | | |
| Male | 0.723 (0.395–1.323) | 0.293 | | |
| Female | Reference | | | |
| Anti-HCV | | | | |
| Positive | 1.758 (0.989–3.126) | 0.054 | | |
| Negative | Reference | | | |
| HBsAg | | | | |
| Positive | 0.498 (0.277–0.895) | 0.020 | 0.569 (0.310–1.046) | 0.069 |
| Negative | Reference | | | |
| AFP (ng/ml) | | | | |
| >20 | 1.345 (0.754–2.400) | 0.316 | | |
| ≦20 | Reference | | | |
| Creatinine (mg/dL) | | | | |
| >1.2 | 2.269 (1.177–4.374) | 0.014 | 1.932 (0.985–3.787) | 0.055 |
| ≦1.2 | Reference | | | |
| Child Pugh class | | | | |
| B | 0.612 (0.084–4.453) | 0.628 | | |
| A | Reference | | | |
| Platelet ($10^9$/L) | | | | |
| <150 | 1.543(0.803–2.965) | 0.193 | | |
| ≧150 | reference | | | |
| Albumin (g/dL) per 1 increase | 0.688(0.373–1.267) | 0.230 | | |
| MELD score ≧9 | 1.341(0.643–2.797) | 0.434 | | |
| MELD score <9 | reference | | | |
| ALBI grade 2 or 3 | 1.251(0.556–2.815) | 0.588 | | |
| ALBI grade 1 | Reference | | | |
| Type of resection | | | | |
| Major | 0.624 (0.310–1.256) | 0.186 | | |
| Minor | Reference | | | |
| The approach of resection | | | | |
| Open | 1.086 (0.531–2.222) | 0.822 | | |
| Laparoscopic | Reference | | | |
| Anatomic resection | | | | |
| No | 0.839 (0.436–1.617) | 0.601 | | |
| Yes | Reference | | | |
| Blood loss during operation | | | | |
| >100ml | 0.812 (0.457–1.444) | 0.479 | | |
| ≦100ml | Reference | | | |
| Operative duration (hours) Per 1 hour increase | 0.830 (0.672–1.026) | 0.085 | | |
| Microvascular invasion | | | | |
| Yes | 1.301 (0.718–2.360) | 0.386 | | |
| No | Reference | | | |
| Cirrhosis | | | | |

(*Continued*)

**Table 3.** (*Continued*)

| | Univariable | | Multivariable | |
|---|---|---|---|---|
| | HR (95%CI) | P | HR (95%CI) | P |
| Yes | 1.584 (0.866–2.897) | 0.135 | | |
| No | Reference | | | |
| Tumor differentiation | | | | |
| Poor | 4.311 (1.325–14.028) | 0.015 | 2.823 (0.842–9.470) | 0.093 |
| Well or moderate | Reference | | | |

MVI, microscopic vascular invasion; AFP, α-feto protein; HBsAg, hepatitis B surface antigen; HCV, hepatitis C virus; MELD, Mayo End-Stage Liver Disease; ALBI, Albumin-Bilirubin

the included patients by using the Cancers Screening and Tracing Information Integrated System for Taiwan (https://hosplab.hpa.gov.tw/CSTIIS/index.aspx). We could thus be sure of the vital status of every single patient enrolled in the study.

There were limitations to the present study. First, this was a retrospective and single institution study, such that the results of the current study may not be generalizable to other institutions. Secondly, the follow-up duration of this study was relatively short compared to previous studies [3, 8, 10–12], and was only able to detect 18 survival events (7.9%) and 45 recurrence events (19.7%). The results of this study should therefore be interpreted with caution.

## Conclusion

The presence of MVI did not impact OS and RFS rates in the patients with solitary HCC up to 2 cm who underwent LR in the present study. The 5-year OS and RFS rates were excellent in this population irrespective of MVI status, which was compatible with the outcomes reported in a previous study [3]. In addition, this study further supports that 8[th] AJCC TNM stage T1a is appropriate for HCC patients undergoing LR.

## Supporting information

**S1 Fig. Algorithm for patient enrollment, 2007–2017.**
(TIFF)

**S2 Fig. Overall survival rates of cirrhotic patients with solitary hepatocellular carcinoma according to presence of microvascular invasion (MVI).**
(TIF)

**S3 Fig. Recurrent-free survival rates of cirrhotic patients with solitary hepatocellular carcinoma according to presence of microvascular invasion (MVI).**
(TIF)

**S4 Fig. Overall survival rates of non-cirrhotic patients with solitary hepatocellular carcinoma according to presence of microvascular invasion (MVI).**
(TIF)

**S5 Fig. Recurrent-free survival rates of non-cirrhotic patients with solitary hepatocellular carcinoma according to presence of microvascular invasion (MVI).**
(TIF)

**S6 Fig. Overall survival rates of female patients with solitary hepatocellular carcinoma according to presence of microvascular invasion (MVI).**
(TIF)

**S7 Fig. Recurrent-free survival rates of female patients with solitary hepatocellular carcinoma according to presence of microvascular invasion (MVI).**
(TIF)

**S8 Fig. Overall survival rates of male patients with solitary hepatocellular carcinoma according to presence of microvascular invasion (MVI).**
(TIF)

**S9 Fig. Recurrent-free survival rates of male patients with solitary hepatocellular carcinoma according to presence of microvascular invasion (MVI).**
(TIF)

**S10 Fig. Overall survival rates of HBsAg positive patients with solitary hepatocellular carcinoma according to presence of microvascular invasion (MVI).**
(TIF)

**S11 Fig. Recurrent-free survival rates of HBsAg positive patients with solitary hepatocellular carcinoma according to presence of microvascular invasion (MVI).**
(TIF)

**S12 Fig. Overall survival rates of anti-HCV positive patients with solitary hepatocellular carcinoma according to presence of microvascular invasion (MVI).**
(TIF)

**S13 Fig. Recurrent-free survival rates of anti-HCV positive patients with solitary hepatocellular carcinoma according to presence of microvascular invasion (MVI).**
(TIF)

**S14 Fig. Overall survival rates of HBsAg positive patients with solitary hepatocellular carcinoma according to use of antiviral therapies.**
(TIF)

**S15 Fig. Recurrent-free survival rates of HBsAg positive patients with solitary hepatocellular carcinoma according to use of antiviral therapies.**
(TIF)

**S16 Fig. Overall survival rates of anti-HCV positive patients with solitary hepatocellular carcinoma according to use of antiviral therapies.**
(TIF)

**S17 Fig. Recurrent-free survival rates of anti-HCV positive patients with solitary hepatocellular carcinoma according to use of antiviral therapies.**
(TIF)

## Acknowledgments

The authors thank Cancer Center, Kaohsiung Chang Gung Memorial Hospital for the provision of HCC registry data. The authors thank Chih-Yun Lin and Nien-Tzu Hsu and the Biostatistics Center, Kaohsiung Chang Gung Memorial Hospital for statistics work. The authors would like to thank Dr. Fang-Ying Kuo, who revised this manuscript critically for important intellectual content.

## Author Contributions

**Conceptualization:** Wei-Feng Li, Chih-Chi Wang, Yi-Hao Yen.

**Data curation:** Wei-Feng Li, Yueh-Wei Liu, Chee-Chien Yong.

**Formal analysis:** Yueh-Wei Liu.

**Methodology:** Yueh-Wei Liu.

**Supervision:** Yueh-Wei Liu, Chih-Chi Wang, Chee-Chien Yong, Chih-Che Lin.

**Validation:** Chih-Chi Wang, Chee-Chien Yong, Chih-Che Lin.

**Visualization:** Chih-Chi Wang, Chee-Chien Yong, Chih-Che Lin.

**Writing – original draft:** Wei-Feng Li, Yi-Hao Yen.

**Writing – review & editing:** Wei-Feng Li, Yi-Hao Yen.

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
