## [Decision Letter · Decision Letter 0]

29 Dec 2022

PONE-D-22-30980Microscopic vascular invasion is not associated with survival in solitary hepatoma ≤ 2 cm underwent resectionPLOS ONE

Dear Dr. Yen,

Thank you for submitting your manuscript to PLOS ONE. After careful consideration, we feel that it has merit but does not fully meet PLOS ONE’s publication criteria as it currently stands. Therefore, we invite you to submit a revised version of the manuscript that addresses the points raised during the review process.

We look forward to receiving your revised manuscript.

Kind regards,

Ibrahim Umar Garzali, MBBS, FWACS

Academic Editor

PLOS ONE

Journal Requirements:

"This study was supported by Grant CMRPG8L0181 from the Chang Gung Memorial Hospital-Kaohsiung Medical Center, Taiwan."

Reviewers' comments:

Reviewer's Responses to Questions

**Comments to the Author**

1. Is the manuscript technically sound, and do the data support the conclusions?

Reviewer #1: Yes

Reviewer #2: No

2. Has the statistical analysis been performed appropriately and rigorously? 

Reviewer #1: Yes

Reviewer #2: Yes

3. Have the authors made all data underlying the findings in their manuscript fully available?

Reviewer #1: No

Reviewer #2: Yes

4. Is the manuscript presented in an intelligible fashion and written in standard English?

Reviewer #1: Yes

Reviewer #2: Yes

5. Review Comments to the Author

Reviewer #1: Thank you for sharing us your valuable study. I have some questions about your manuscript. Please find them below.

- Firstly, why did not you prefer local ablative treatment firstly in your study? Because the sizes of the tumors were appropriate for ablative treatment.

- Secondly, why did you choose major resection for some patients? As you know, major resection is unexpected result for these size of HCC.

- Thirdly what is AFP upper limit in this study and ecog status of patients?

Reviewer #2: This is a reasonable study based on a small dataset, as the authors acknowledge. They did not find and significant differences in survival parameters for OS or for the larger cohort of RFS. Therefore it is unsurprising that they found no effect for MVI. That they did not find an effect does not mean there is no effect. Therefore the title should reflect this. They cannot claim there is no effect of MVI, merely that they could not find an effect of MVI - or anything else- on OS, probably due to small cohort size, but likely because of the high OS, regardless of MVI status. As such, this is a small contribution to the literature.

6. PLOS authors have the option to publish the peer review history of their article (what does this mean?). If published, this will include your full peer review and any attached files.

Reviewer #1: No

Reviewer #2: No

---

## [Author Response · Author response to Decision Letter 0]

10 Jan 2023

Reviewer #1: Thank you for sharing us your valuable study. I have some questions about your manuscript. Please find them below.

- Firstly, why did not you prefer local ablative treatment firstly in your study? Because the sizes of the tumors were appropriate for ablative treatment.

Response: Thank you for your comments. Our previous study enrolled 143 patients with Barcelona Clinic Liver Cancer (BCLC) very early-stage HCC; of these patients, 52 underwent liver resection (LR) and 91 underwent radiofrequency ablation (RFA). The 3- and 5-year overall survival (OS) rates were 98% and 91.5% for LR and 80.3% and 72% for RFA, respectively (p = 0.073). The 3- and 5-year recurrence-free survival (RFS) rates were 62.1% and 40.7% for LR and 39.8% and 29.3% for RFA, respectively (p = 0.006). We concluded that for patients with Barcelona clinic liver cancer (BCLC) very early-stage HCC, there was no significant difference in OS between LR and RFA. However, LR resulted in better RFS than RFA [1]. Therefore, based on the results of our previous study [1], we recommend LR for patients with BCLC very early-stage HCC if they are good candidates for surgery, i.e., they have good liver function reserve and performance status and do not have severe comorbidities, and RFA as an alternative treatment modality. 

Reference 

1. Wang JH, Wang CC, Hung CH, Chen CL, Lu SN. Survival comparison between surgical resection and radiofrequency ablation for patients in BCLC very early/early stage hepatocellular carcinoma. J Hepatol. 2012;56:412-8. doi: 10.1016/j.jhep.2011.05.020. 

- Secondly, why did you choose major resection for some patients? As you know, major resection is unexpected result for these size of HCC.

Response: As mentioned in the European Association for the Study of the Liver (EASL) guidelines, RFA offers comparable results to those of LR for deeply/centrally located single HCC of ≤ 2 cm. In contrast, laparoscopic–robotic LR for HCC located in superficial/peripheral sites of the liver provides optimal survival outcomes, while minimizing complications and hospital stay [1]. However, based on the results of our previous study [2], we recommend LR for patients with BCLC very early-stage HCC, even for a tumor that is deeply/centrally located and might require major resection. 

References 

1. European Association for the Study of the Liver. EASL Clinical Practice Guidelines: Management of hepatocellular carcinoma. European Association for the Study of the Liver. J Hepatol. 2018;69:182-236. doi: 10.1016/j.jhep.2018.03.019.

2. Wang JH, Wang CC, Hung CH, Chen CL, Lu SN. Survival comparison between surgical resection and radiofrequency ablation for patients in BCLC very early/early stage hepatocellular carcinoma. J Hepatol. 2012;56:412-8. doi: 10.1016/j.jhep.2011.05.020. 

- Thirdly what is AFP upper limit in this study and ecog status of patients? 

Response: 

1. Our institution uses a chemiluminescent microparticle immunoassay (CMIA), manufactured by Abbott, for quantitative determination of alpha-fetoprotein (AFP). The AFP upper limit is 80,000 ng/ml. 

2. In our institution, LR is recommended for patients who are good candidates for surgery, i.e., those with an ECOG performance status = 0. All patients in the present study had an ECOG performance status = 0. Patients with inadequate PS would be referred for RFA. 

Reviewer #2: This is a reasonable study based on a small dataset, as the authors acknowledge. They did not find and significant differences in survival parameters for OS or for the larger cohort of RFS. Therefore it is unsurprising that they found no effect for MVI. That they did not find an effect does not mean there is no effect. Therefore the title should reflect this. They cannot claim there is no effect of MVI, merely that they could not find an effect of MVI - or anything else- on OS, probably due to small cohort size, but likely because of the high OS, regardless of MVI status. As such, this is a small contribution to the literature.

Response: Thank you for your comments. We have revised the title to “Microscopic vascular invasion may not be associated with survival of patients undergoing resection for solitary hepatoma of ≤ 2 cm.”

---

## [Editor Report · Decision Letter 1]

16 Jan 2023

Microscopic vascular invasion may not be associated with survival of patients undergoing resection for solitary hepatoma of ≤ 2 cm”

PONE-D-22-30980R1

Dear Dr. Yen,

We’re pleased to inform you that your manuscript has been judged scientifically suitable for publication and will be formally accepted for publication once it meets all outstanding technical requirements.

Kind regards,

Ibrahim Umar Garzali, MBBS, FWACS

Academic Editor

PLOS ONE